# Drug Inhibition of Redox Factor-1 Restores Hypoxia-Driven Changes in Tuberous Sclerosis Complex 2 Deficient Cells

**DOI:** 10.3390/cancers14246195

**Published:** 2022-12-15

**Authors:** Jesse D. Champion, Kayleigh M. Dodd, Hilaire C. Lam, Mohammad A. M. Alzahrani, Sara Seifan, Ellie Rad, David Oliver Scourfield, Melissa L. Fishel, Brian L. Calver, Ann Ager, Elizabeth P. Henske, David Mark Davies, Mark R. Kelley, Andrew R. Tee

**Affiliations:** 1Division of Cancer and Genetics, Cardiff University, Heath Park, Cardiff CF14 4XN, UK; 2Pulmonary and Critical Care Medicine, Department of Medicine, Brigham and Women’s Hospital, Harvard Medical School, Boston, MA 02115, USA; 3Herman B Wells Center for Pediatric Research, Indiana University School of Medicine, Indianapolis, IN 46202, USA; 4Division of Infection and Immunity, Cardiff University, Heath Park, Cardiff CF14 4XN, UK; 5Department of Oncology, South West Wales Cancer Centre, Singleton Hospital, Swansea SA2 8QA, UK

**Keywords:** TSC, Ref-1, APE1, mTOR, HIF-1α, STAT3, NF-kB, angiogenesis, hypoxia, redox

## Abstract

**Simple Summary:**

Tuberous sclerosis complex (TSC) is a genetic disease where patients are predisposed to tumors and neurological complications. Current therapies for this disease are not fully curative. We aimed to explore novel drug targets and therapies that could further benefit TSC patients. This work uncovered a novel pathway that drives disease in TSC cell models involving redox factor-1 (Ref-1). Ref-1 is a protein that turns on several key transcription factors that collectively promote tumor growth and survival through direct redox signaling. Processes regulated by Ref-1 include angiogenesis, inflammation, and metabolic transformation. Therefore, this work reveals a new drug target, where inhibitors of Ref-1 could have an additional benefit compared to current drug therapies.

**Abstract:**

Therapies with the mechanistic target of rapamycin complex 1 (mTORC1) inhibitors are not fully curative for tuberous sclerosis complex (TSC) patients. Here, we propose that some mTORC1-independent disease facets of TSC involve signaling through redox factor-1 (Ref-1). Ref-1 possesses a redox signaling activity that stimulates the transcriptional activity of STAT3, NF-kB, and HIF-1α, which are involved in inflammation, proliferation, angiogenesis, and hypoxia, respectively. Here, we demonstrate that redox signaling through Ref-1 contributes to metabolic transformation and tumor growth in TSC cell model systems. In TSC2-deficient cells, the clinically viable Ref-1 inhibitor APX3330 was effective at blocking the hyperactivity of STAT3, NF-kB, and HIF-1α. While Ref-1 inhibitors do not inhibit mTORC1, they potently block cell invasion and vasculature mimicry. Of interest, we show that cell invasion and vasculature mimicry linked to Ref-1 redox signaling are not blocked by mTORC1 inhibitors. Metabolic profiling revealed that Ref-1 inhibitors alter metabolites associated with the glutathione antioxidant pathway as well as metabolites that are heavily dysregulated in TSC2-deficient cells involved in redox homeostasis. Therefore, this work presents Ref-1 and associated redox-regulated transcription factors such as STAT3, NF-kB, and HIF-1α as potential therapeutic targets to treat TSC, where targeting these components would likely have additional benefits compared to using mTORC1 inhibitors alone.

## 1. Introduction

Tuberous sclerosis complex (TSC) is a rare autosomal dominant genetic condition caused by inactivating mutations in either the *TSC1* or *TSC2* genes. TSC patients are predisposed to benign tumors in multiple organs, including the kidneys, skin, brain, and heart (for reviews, see [1,2]). Renal angiomyolipomas (AMLs) are the most frequent tumors, affecting approximately 80% of individuals with TSC, and are a leading cause of morbidity in adulthood [3]. AMLs are highly vascularized and are composed of adipose and smooth muscle tissue. About 90% of TSC patients also have at least one skin lesion that develops early in life. Facial angiofibromas are the most visible cutaneous feature of TSC and consist of fibrous tissue and blood vessels. Brain tumors are less frequent, but those that occur can be associated with epilepsy. However, the formation of brain tumors is not necessarily required for the development of epilepsy, where approximately 90% of TSC patients experience a seizure during their lifetime [4]. A proportion of seizures are resistant to antiepileptic medications, making them difficult to control. Lymphangioleiomyomatosis (LAM), the pulmonary manifestation of TSC, almost exclusively affects women. TSC2-deficient LAM cells are believed to metastasize to the lungs, resulting in cystic lung destruction [5,6]. The pathophysiology of TSC is complex. While we have made significant forward progress, there is much we still need to understand. A deeper knowledge of the signaling mechanism defects that contribute to disease development is required and will allow the development of better therapeutic strategies to treat the disease manifestations of TSC.

A key feature of TSC biology is the ability of the TSC1/TSC2 tumor suppressor protein complex to inhibit the mechanistic target of the rapamycin complex 1 (mTORC1) signaling pathway involved in normal cell growth control. The small G protein Ras homologue enriched in brain (Rheb) is negatively regulated by the GTPase-activating protein (GAP) domain of TSC2 [7]. Loss-of-function *TSC1* or *TSC2* mutations constitutively switch Rheb to an active GTP-bound state that then aberrantly activates mTORC1, leading to metabolic transformation and uncontrolled tumor growth and encouraging angiogenesis in hypoxic tissues (reviewed in [8]). The fundamental discovery of the Rheb-GAP tumor suppressor function of TSC2 positioned mTORC1 as a new drug target for the treatment of TSC. Consequently, rapamycin analogues (rapalogues) that inhibit mTORC1 are now clinically approved for the treatment of TSC. While improving the quality of life for TSC patients, mTORC1 inhibition does not completely restore the disease state. Data from both in vitro and clinical studies provide evidence that rapamycin is a cytostatic agent and that the cessation of therapy results in the rebound and regrowth of TSC tumors [9]. Further investigation into the disease mechanisms of TSC is clearly required to reveal new drug targets and to advance therapeutic options.

While mTORC1 hyperactivation upon the loss of *TSC1*/*TSC2* is a well-defined aspect of TSC that is mechanistically linked to tumor growth, we hypothesize that mTORC1-independent signaling mechanisms are also critically involved in the development of TSC. mTORC1-independent mechanisms might be involved in the disease progression of TSC and LAM and could help explain why mTORC1 inhibitors only have a partial but stable response in TSC/LAM patients. We speculate that the hypoxic signaling involved in angiogenesis and tumor growth becomes elevated in hypoxic TSC2-deficient tissue that is not dependent on mTORC1. Research has indicated that hypoxic signaling is a key driver of TSC [10,11], where the loss of *TSC2* can pathologically promote the expression of hypoxia inducible factor 1α (HIF-1α). HIF-1α regulation is multifaceted, with multiple inputs that regulate its mRNA/protein expression, transcriptional activity, and protein stability. We previously discovered that signal transducer and activator of transcription 3 (STAT3) functions upstream of HIF-1α during hypoxia in TSC2-deficient cell models [11] and is required for HIF-1α mRNA gene expression. STAT3 has previously been shown to have an elevated level of Tyr705 phosphorylation in TSC2-deficient models [12,13,14]. Tyr705 phosphorylation is required for STAT3 activation and is regulated through tyrosine kinases such as Janus kinase 2 (JAK2). Nuclear factor kappa-light-chain-enhancer of activated B cells (NF-κB) is another potential transcriptional input to HIF-1α [15] and to date has been understudied in TSC. HIF-1α activation leads to angiogenesis and tumor development, key features that are intrinsically linked to TSC. We postulate that TSC-associated pathology involves STAT3, NF-κB, and HIF-1α.

We speculated that this STAT3, NF-κB, and HIF-1α transcriptional signaling nexus might play a role in the disease progression of TSC and LAM. Given that mTORC1 inhibitors are not fully restorative, we investigated another potential therapeutic strategy to restore the disease features of TSC2-deficient cells. Through this work, we revealed a novel drug target for the therapy of TSC called Ref-1 (also known as APE1 or APEX1). Ref-1 is involved in redox signaling pathways associated with cancer, including angiogenesis, proliferation, inflammation, and the hypoxia response (reviewed in [16]).

## 2. Materials and Methods

### 2.1. Antibodies and Biochemicals

Anti-TSC2, anti-STAT3 phospho-Ser727 and Tyr705, Ref-1, Rel-A phospho-Ser365, anti-β-actin, VEGF-A, BNIP3, and anti-rpS6 phospho-Ser235/236 were obtained from Cell Signaling Technology (Danvers, MA, USA). Anti-HIF-1α was purchased from BD Transduction Laboratories (Oxford, UK). Rapamycin, FL3331, and JSH23 were bought from Merck (Darmstadt, Germany), and KU-0063794 was purchased from Chemquest Ltd. (Cheshire, UK), while APX3330, APX2009, APX2011, and RN7-58 were obtained from Apexian Pharmaceuticals (Indianapolis, IN, USA). Unless stated otherwise, all other lab chemicals were obtained from Merck.

### 2.2. Cell Culture, Transfection, and Preparing Cell Lysates

*Tsc2*+/+*Tp53*−/− and *Tsc2*−/−*Tp53*−/− mouse embryonic fibroblasts (MEFs) were a kind gift from Prof. D. Kwiatkowski (Harvard University, Boston, MA, USA) and were cultured in Dulbecco’s modified Eagle’s medium supplemented with 10% (*v*/*v*) fetal calf serum and 100 μ/mL penicillin streptomycin (Life Technologies/ThermoFisher Scientific, Waltham, MA, USA). Human TSC2-deficient angiomyolipoma AML 621-101 cells [17] were grown as above but were supplemented with 20% (*v*/*v*) fetal calf serum. Transfections were carried out using a JETPei transfection reagent (VWR International, Leicestershire, UK) as directed by the manufacturer’s protocol. Drugs were treated as indicated. Hypoxia experiments were set up in a Binder CB150 hypoxic chamber set at 1% O_2_ overnight (18 h). Cells were lysed directly in sample buffer (62.5 mM Tris–HCl (pH 7.6), 50 mM dithiothreitol, 2% (*w*/*v*) sodium dodecyl sulfate, 10% (*w*/*v*) glycerol, and 0.1% (*w*/*v*) bromophenol blue), sonicated for three 20 s cycles on full power (30 amplitude microns), heated at 95 °C for 10 min, and centrifuged at 13,000 rpm for 8 min. Protein quantification was carried out using the Pierce 660 nm protein reagent (with an ionic detergent compatibility reagent bought from ThermoFisher Scientific, Waltham, MA, USA) before being subjected to Western blotting.

### 2.3. Western Blotting

Cell protein lysates and immunoprecipitated protein samples were subjected to sodium dodecyl sulfate–polyacrylamide gel electrophoresis using 12-lane NuPage precast gels (Life Technologies/ThermoFisher Scientific, Waltham, MA, USA). Proteins were transferred to polyvinylidene difluoride membranes (Millipore, Edinburgh, UK), which were then blocked in 5% (*w*/*v*) dry milk powder in Tris-buffered saline with 0.1% (*v*/*v*) Tween. Blots were probed sequentially with primary antibodies overnight (1:1000) and then with horse radish peroxidase conjugated secondary antibodies (1:10,000) with washing with Tris-buffered saline with 0.1% (*v*/*v*) Tween. An enhanced chemiluminescent solution was used to detect protein bands with hyperfilm (both GE Healthcare, Buckinghamshire, UK). All Western blots are representative of at least three independent experiments. The processing of images was carried out using ImageJ (v.50), where background levels were removed by a small adjustment in the contrast and brightness levels.

### 2.4. Anchorage-Independent Cell Growth in Soft Agar

Six-well plastic tissue culture plates were coated with melted 0.6% (*w*/*v*) agar in phosphate-buffered saline. On top of this bottom layer, after setting at room temperature, either MEFs (50,000 cells) or AML 621-101 (12,000 cells) were resuspended as single cells in melted 0.3% (*w*/*v*) agar and phosphate-buffered saline with 50% (*v*/*v*) Dulbecco’s modified Eagle’s medium supplemented at 40 °C. The top agar/cell layer was set in a 37 °C humidified tissue culture incubator. The working concentrations of each drug were set up within each layer prior to plating. Next, 2 mL of medium was added with working concentrations of each drug or DMSO (vehicle). Plates were grown for 4 weeks, with the medium and drug replaced twice weekly, before pictures were taken. The colony diameter was determined using ImageJ (v.50), and the colony numbers were scored.

### 2.5. Vasculature Mimicry

A 96-well tissue culture plate was coated with 50 µL of a chilled Matrigel (growth-factor-reduced and LDEV-free) basement membrane matrix (Corning Inc., New York, NY, USA) and set at room temperature. Either 30,000 *Tsc2*−/− MEF cells or 50,000 AML 621-101 cells were resuspended in Optimem reduced serum media (Life Technologies, Thermo Fisher Scientific, Waltham, MA, USA) and added to each coated well. Cells were treated with the specified drugs or vehicle only and then placed in a hypoxic incubator. After 16 h, pictures were taken on an EVOS XL Core camera and analyzed in ImageJ (v.50) using the AngioTool software.

### 2.6. Cytotoxicity Assays: Assays for Spheroids, Outgrowth, and Acridine Orange/Propidium Iodide (AO/PI) Cell Viability

Spheroid formation and outgrowth analyses were performed as previously described [18]. Spheroids and cell outgrowth were imaged using an EVOS XL Core camera and then analyzed using ImageJ (v.50). Cell viability was assessed using an AO/PI stain reagent following the manufacture’s guidelines (Labtech, Healthfield, UK). Cells mixed with AO/PI were analyzed on a LUNA-FL™ Dual Fluorescence Cell Counter.

### 2.7. Transcription Assays

Inducible luciferase reporter constructs for HIF-1α, STAT3, and NF-κB were purchased from Affymetrix Inc. (Milan, Italy). Transfections were carried out using JETPei (VWR International, Lutterworth, UK) according to the manufacturer’s protocol. A Promega dual-luciferase reporter assay system was used, in accordance with the manufacturer’s protocol (Promega, Southampton, UK), using 20 µL of protein lysate. The luminescence was measured and standardized for the protein content.

### 2.8. Targeted Metabolic Profiling

To account for differences in the cell size and protein content at harvest, *Tsc2*+/+ (8 × 10^5^) and *Tsc2*−/− (4 × 10^5^) MEFs were differentially seeded onto 10 cm^2^ plates. The next day, the cells were changed to 0.1% (*v*/*v*) FBS DMEM, and vehicle or 25 μM APX3330 were added ~6 h later. After overnight treatment, metabolites were extracted with ice-cold high-performance liquid chromatography (HPLC)-grade 80% (*v*/*v*) aqueous methanol from five replicate 10 cm^2^ plates on dry ice, as described previously [19]. Metabolites were processed and analyzed using selected reaction monitoring (SRM) with polarity switching on a 5500 QTRAP triple quadrupole mass spectrometer (AB/SCIEX) coupled to a Prominence UFLC HPLC system (Shimadzu) using amide hydrophilic interaction chromatography at pH 9.2. Then, 296 endogenous water-soluble metabolites were measured in a steady state (Metabolomics Core, Beth Israel Deaconess Medical Center, Boston, MA, USA). The resulting raw metabolomics data were normalized to the protein concentrations of three additional replicate plates by multiplying a correction factor for each treatment group (calculated as the mean total protein of all samples/the mean total protein of the group) and uploaded into MetaboAnalyst 3.0 (http://www.metaboanalyst.ca/MetaboAnalyst/, accessed on 1 November 2018) for subsequent processing and analysis. The peak area intensities for each metabolite were then normalized relative to vehicle-treated *Tsc2*+/+ MEFs. Specifically, to generate processed data, the peak areas were filtered by interquartile range, autoscaled (mean-centered and divided by the standard deviation for each variable), and Log^2^ transformed, and a statistical analysis was applied to generate heatmaps and a principal component analysis in MetaboAnalyst 3.0.

### 2.9. Oxygen Consumption and Extracellular Acidification Rate

Cells were seeded at 1 × 10^4^ cells per well in 100 μL of 10% (*v*/*v*) FBS DMEM and allowed to attach to the bottom of the assay plate. The cells were treated with vehicle or 2 μM APX3330 ~6 h later in 500 μL of 10% (*v*/*v*) FBS DMEM. The next day, the basal oxygen consumption rate and basal extracellular acidification rate were measured using a Seahorse Bioscience XF24 analyzer (Agilent, Santa Clara, CA, USA). The data were normalized by fixing the wells and staining with DAPI to quantify the total nuclei per well.

### 2.10. Experimental Repeats and Statistical Analysis

The results are representative of three experiments unless otherwise stated. The statistical analysis was carried out using GraphPad Prism 9, using multiple unpaired *t*-tests or an ANOVA when multiple comparisons were being carried out. A one-way ANOVA with Tukey’s post hoc test was used, unless the data were non-parametric and were instead analyzed with a Kruskall–Wallis test and Dunn’s post hoc test. In the figures, *p*-values are represented as <0.05 *, <0.01 **, <0.001 ***, or not significant ‘NS’.

## 3. Results

### 3.1. Aberrant Activation of HIF-1α/STAT3/NF-κB in Tsc2-Deficient Cells Is Restored with APX3330, a Ref-1-Specific Inhibitor

Initially, we compared STAT3 phosphorylation at Y705 and the protein levels of HIF-1α in *Tsc2*+/+ and *Tsc2*−/− MEFs under normoxia and hypoxia (Figure 1a). We observed a marked increase in the Y705 phosphorylation of STAT3 and HIF-1α protein in the *Tsc2*−/− MEFs under hypoxia when compared to the wild-type *Tsc2*+/+ MEFs, showing that hypoxia causes an aberrant upregulation of these transcription factors in the absence of *Tsc2*. The STAT3 inhibitor FLLL31 inhibited the hypoxic induction of STAT3 Y705 phosphorylation but not the accumulation of HIF-1α protein. While it is known that the loss of *TSC2* activates STAT3 [12,13,14], the degree of STAT3 hyperactivity in *TSC2*-deficient cells has not been quantified to date. Therefore, we carried out transcription assays on STAT3. We observed a marked elevation in the transcription activity of STAT3 (Figure 1b) in *Tsc2*−/− MEFs under hypoxia: an over 3-fold induction when compared to the *Tsc2*+/+ wild-type controls. Similarly, the activity of another hypoxia-responsive transcription factor, NF-κB, was markedly upregulated in the *Tsc2*−/− MEFs under hypoxia when compared to wild-type *Tsc2*+/+ MEFs (Figure 1c). Our data indicate that hypoxic responsive transcription factors such as HIF-1α, STAT3, and NF-κB become aberrantly activated when TSC2 is deficient.

Tumor formation assays were carried out in angiomyolipoma (AML) cells from an LAM patient (621-102) lacking functional TSC2 (Figure 1d). We found a significant reduction in tumor diameter with the ATP-competitive mTOR inhibitor (Ku-0063794), as expected. We also observed a significant reduction in tumor diameter with inhibitors of STAT3 and NF-κB (FLLL31 and JSH23, respectively), showing that these transcription factors are likely involved in tumor growth.

Given the above experimental evidence, we sought to find a therapeutic strategy where a single drug treatment might restore the activity of the three key hypoxia-responsive transcription factors STAT3, NF-κB, and HIF-1α in TSC2-deficient cells. Fortuitously, Ref-1 is upstream of these transcription factors and has been shown to regulate their DNA-binding activities through its redox function (reviewed in [16]). Ref-1 has two key functions in mammalian cells: DNA base excision repair activity and redox signaling. During conditions of redox stress, Ref-1 enhances the DNA-binding activity of multiple transcription factors, including HIF-1α, STAT3, and NF-κB, through a reduction in critical cysteine residues within their DNA-binding or transactivation domains. To investigate the role that Ref-1 might play in promoting HIF-1α, STAT3, and NF-κB during hypoxia-mediated oxidative stress in TSC2, we first examined the levels of Ref-1 protein. We observed at least 2-fold higher expression of Ref-1 in *Tsc2*−/− MEFs when compared to wild-type controls (Figure 1e). To examine the effects of Ref-1 inhibition on these transcription factors in TSC, *Tsc2*−/− MEFs were treated with the specific Ref-1 inhibitor APX3330, and the effects on the transcriptional activity of HIF-1α (Figure 2a), STAT3 (Figure 2b), and NF-κB (Figure 2c) were measured under hypoxia (1% O_2_). Of importance, APX3330 at 100 µM restored the aberrant activation of HIF-1α, STAT3, and NF-κB to a level comparable to cells re-expressing TSC2. At the lower 50 µM concentration, APX3330 treatment resulted in a significant reduction in the activity of HIF-1α, STAT3, and NF-κB.

Under hypoxia, 50 µM APX3330 was sufficient to reduce HIF-1α protein expression and the downstream HIF-1α targets VEGF-A and BNIP3 (Figure 2d). Unlike mTOR inhibition with either rapamycin or KU-006379, Ref-1 inhibition with APX3330 did not reduce ribosomal protein S6 (rpS6) phosphorylation. Therefore, the drug action for APX3330 to inhibit HIF-1α/VEGF-A/BNIP3 signaling is mTORC1-independent. Similar to rapamycin, APX3330 did not reduce STAT3-Y705 or RelA-S276 phosphorylation (Appendix A), indicating that APX3330 inhibits the DNA binding of these transcription factors, as is expected based on the redox function of Ref-1 [20]. Ref-1 presents as a new drug target for TSC (depicted in Figure 3), where Ref-1 inhibition functions as a different restorative therapy to the currently used mTOR inhibitors.

### 3.2. APX3330 Inhibits Cell Migration/Invasion and Tumor Spheroid Formation of Tsc2−/− MEFs

To explore whether Ref-1 inhibition has potency to revert other TSC-associated disease features, we carried out cell migration/invasion and tumor growth assays in *Tsc2*−/− MEFs after APX3330 treatment. At doses where HIF-1α, STAT3, and NF-κB are inhibited, APX3330 impaired the wound closure of *Tsc2*−/− MEFs when compared to vehicle (Figure 4a). Furthermore, both cell migration (Figure 4b) and invasion (Figure 4c) were markedly impaired by increasing amounts of APX3330. As expected, the migration (Figure 4b) and invasion (Figure 4c) of the *Tsc2*+/+ MEFs were significantly lower when compared to the *Tsc2*−/− MEFs. While APX3330 treatment was effective at reducing this high level of cell migration and invasion in the *Tsc2*−/− MEFs, rapamycin treatment was less effective. We next assessed the in vitro tumor growth of *Tsc2*−/− MEFs when grown in soft agar in the presence or absence of APX3330 (Figure 4d). We observed a marked reduction in the diameter of tumors in the presence of APX3330 (at 25, 50, and 100 μM). The data reveal that Ref-1 redox-mediated signaling is a driver of cell migration, invasion, and tumor growth in *TSC2*-deficient cells.

### 3.3. Metabolic Profiling Shows Restoration of Redox Homeostasis in the Tsc2−/− MEFs after APX3330 Treatment

Metabolic profiles were determined in untreated *Tsc2*+/+ and *Tsc2*−/− MEFs, and APX3330-treated *Tsc2*−/− MEFs. Untreated *Tsc2*−/− MEFs were normalized to the untreated *Tsc2*+/+ controls, and key changes in the metabolic profile between the untreated *Tsc2−/−* MEFs and normal untreated wild-type *Tsc2*+/+ MEFs are shown in Figure 5a. In the untreated *Tsc2*−/− MEFs, the Krebs cycle metabolite α-ketoglutarate was significantly higher when compared to *Tsc2*+/+ controls (elevated by 1.6-fold, Figure 5b). *Tsc2*−/− MEFs also had an elevated level of ribose-5-phosphate within the pentose phosphate pathway (PPP, elevated by 1.5-fold in the untreated *Tsc2*−/− MEFs versus the wild-type controls), which is in line with previous reports that the PPP and de novo pyrimidine synthesis are upregulated in TSC2-deficient cells [21]. We also observed that untreated *Tsc2*−/− MEFs had significantly higher levels of glutamine, glutamate, and N-acetyl-glutamate compared to wild-type controls. Of interest, the APX3330 treatment was sufficient to reduce glutamine, glutamate, and N-acetyl-glutamate in the *Tsc2*−/− MEFs. Glutamate and cysteine are essential metabolites in the generation glutathione (GSH), a tripeptide thiol antioxidant that was markedly elevated in the *Tsc2*−/− MEFs (1.9-fold increase) compared to wild-type controls. In cells, GSH is critically involved in antioxidant defense and is typically involved in the reduction of H_2_O_2_ to generate H_2_O and oxidized glutathione disulfide. The differential levels of GSH and glutathione disulfide in the untreated *Tsc2*−/− suggest oxidative stress. Other metabolites that can be biosynthetically derived from the amino acid glutamate were also observed to be elevated in the *Tsc2*−/− MEFs (when compared to wild-type controls), including N-acetyl-glutamate (2.2-fold), ornithine (3.9-fold), proline (2-fold), and putrescine (3.9-fold). These data imply that TSC2-deficient cells have defects in proline synthesis. Supporting this observation, previous work has shown that TSC patients have a higher level of hydroxyproline in their tumors and urine [22] and a higher level of free proline in their blood [23].

A heatmap comparing the three samples, wild-type *Tsc2*+/+, untreated *Tsc2*−/−, and APX3330-treated *Tsc2*−/−, is shown in Appendix A (normalized data, *n* = 5, can be found in Appendix A). Key differences between the untreated and APX3330-treated *Tsc2*−/− MEFs are graphed in Figure 4b. The APX3330 treatment reduced the levels of ribose-5-phosphate and Krebs cycle intermediates, citrate, cis-aconitate, D-isocitrate, and α-ketoglutarate. Previous work showed that the inhibition of Ref-1 redox activity blocked the cells’ ability to utilize the Krebs cycle substrates α-ketoglutarate, succinate, fumarate, and malate [24]. Of interest, APX3330 dramatically reduced the levels of GSH and increased the levels of glutathione disulfide and cysteine, which implies that the APX3330 treatment impedes the generation of GSH and alters redox homeostasis. Ref-1 was previously described as having a redox chaperone activity, where Ref-1 regulated the DNA-binding activity of NF-κB through promoting a reduction in the critical cysteine residues within NF-κB by other reducing molecules such as GSH and thioredoxin (Trx) [25]. Cysteine and GSH function as two major cellular thiol antioxidants. Another potent thiol antioxidant and downstream metabolite of cysteine is the semi-essential non-proteinogenic amino acid taurine, which was also observed to be significantly elevated in the *Tsc2*−/− MEFs when compared to the wild-type controls, and taurine was further enhanced with APX3330. The upstream metabolite of cysteine, cystathionine, was upregulated in the *Tsc2*−/− MEFs (by 10.7-fold) and was reduced with the APX3330 treatment. Other metabolites within the cysteine–methionine pathway, 5-methyl THF and S-adenosyl methionine, were also elevated in the *Tsc2*−/− MEFs and were reduced with APX3330. Proline and cysteine synthesis pathway intermediates appear to be favored in the Tsc2−/− MEFs, which could help explain why there is an elevated pool of reduced glutathione (GSH) (Figure 5a). GSH is a critical scavenger of reactive oxygen species (ROS). Consequently, the ratio of GSH with oxidized glutathione disulfide can be used as a marker of oxidative stress. The ratio of GSH/GSH disulfide was dramatically altered with APX3330 treatment (Figure 6a, with a ratio of 1:12 with APX3330). This shows that APX3330 caused oxidative stress to the Tsc2−/− MEFs.

We next examined the oxygen consumption (Figure 6b) and extracellular acidification rates (Figure 6c). While we observed that APX3330 did not change the oxygen consumption rates of the *Tsc2*−/− MEFs, the APX3330 treatment dramatically reduced the extracellular acidification rates. Extracellular acidification is linked to the HIF-1α activation and lactic acid secretion that occurs during non-aerobic respiration. These results are in line with previous research where APX3330 was found to reduce both HIF-1α activity via carbonic anhydrase IX (CAIX) and intracellular pH in pancreatic cancer cells [26]. In summary, APX3330 treatment reduced non-aerobic respiration, restored glutamine–glutamate homeostasis, and altered the balance of antioxidant metabolites linked to the regeneration of GSH in the *Tsc2*−/− MEFs.

### 3.4. Second-Generation Ref-1 Inhibitor APX2009 Shows Increased Potency to Inhibit HIF-1α

Compared to APX3330, the second-generation inhibitor APX2009 has increased potency to inhibit Ref-1 in various cancer cell models [26,27]. We compared the drug activity of rapamycin, APX3330, and APX2009 alone and in combination on HIF-1α activity by carrying out HIF-1α transcriptional activity assays in *Tsc2*−/− MEFs subjected to hypoxia (Figure 7). At 100 µM, APX3330 inhibited HIF-1α by 90%, while rapamycin only reduced the activity of HIF-1α by 40%. The combination of APX3330 at 25 µM with rapamycin showed a greater inhibition of HIF-1α when compared to single-drug treatments. As expected, we observed greater potency to inhibit HIF-1α with APX2009 when compared to treatment with the parental drug APX3330 or to rapamycin alone. At 10 μM, APX2009 restored HIF-1α activity to a level equivalent to TSC2 re-expression. We observed no additional effects on HIF-1α activity when rapamycin was combined with APX2009. A treatment with RN7-58 (at 100 μM) was used as a negative control drug, as this modified APX2009 analogue is unable to act as a Ref-1 redox inhibitor [28]. As expected, RN7-58 did not inhibit the hypoxic induction of HIF-1α activity.

### 3.5. Second-Generation Ref-1 Inhibitors APX2009 and APX2014 Inhibit In Vitro Tumor Growth of AML (621-102) Cells

We next wanted to assess the antitumor activity of Ref-1 inhibitors in an LAM patient angiomyolipoma derived cell line (621-102 AML cells) that lacks functional TSC2. We compared the activity of APX3330 with two second-generation inhibitors, APX2009 and APX2014, mTOR inhibitors, rapamycin, and Ku-0063794. We grew 621-102 AML cells in soft agar in the presence and absence of drugs and determined the diameters of the tumor spheroids. Both mTOR inhibitors, rapamycin and Ku-0063794, were effective at reducing tumor diameters (Figure 8a). Upon the comparison of Ku-0063794 and rapamycin, the number of tumor spheroids was decreased after treatment with Ku-0063794, indicating that a more complete inhibition of mTOR with the ATPase-competitive inhibitor prevents colony formation. The 621-102 AML cells require 15% (*v*/*v*) FBS to form colonies in these in vitro tumor formation assays in soft agar. We saw that APX3330 was less effective at reducing the diameters of tumor spheroids (Appendix A), which likely reflects the higher levels of serum used in these assays. In line with this result, we observed less effectiveness of APX3330 with increasing serum concentrations (data not shown) due to the binding of APX3330 to serum proteins, impacting its activity. However, in the presence of 15% (*v*/*v*) FBS, the second-generation inhibitors APX2009 (at 10 and 20 μM) and APX2014 (at 5, 10, and 20 μM) showed enhanced potency to inhibit the number of tumor spheroids that grew (Figure 8a). With APX2009 and APX2014 treatments, there was a significant reduction in colony number. To further confirm the inhibition of HIF-1α, the total protein levels of HIF-1α and its downstream target BNIP3 were analyzed in 621-102 AML cells and *Tsc2*−/− MEFs (Figure 8b). Second-generation Ref-1 inhibitors (APX2009 and APX2014) were effective at reducing either HIF-1α or BNIP3 protein levels, while APX3330 was not, again demonstrating increased potency with new analogues. As previously observed in the *Tsc2*−/− MEFs, the Ref-1 inhibitors did not inhibit mTORC1 in the 621-102 AML cells, as observed by the high levels of rpS6-P after treatment.

To determine whether the drugs had cytotoxic activity, we treated cell spheroids grown on soft agar (*Tsc2*−/− MEFs and 621-102 AML cells) with DMSO, APX3330 (50 or 100 μM), APX2009 (5 or 10 μM), APX2014 (5 or 10 μM), or a known cytotoxic agent, etoposide (10 μM), for 11 days in the presence of serum. We then examined whether cells recovered from treatment by transferring the cell spheroids to standard plastic-coated tissue culture plates in the absence of drugs. Both the *Tsc2*−/− MEFs and 621-102 AML cells recovered after treatment with APX3330, APX2009, or APX2014 (similar to the DMSO control), as observed by cells migrating out of the adhered spheroid within a 48 h period (Appendix A). No migrating cells were observed after the etoposide treatment, showing cytotoxic activity as expected. Acridine orange/propidium iodide cell viability assays showed no loss of cell viability with APX3330 (50 or 100 μM), APX2009 (5 μM), or APX2014 (5 μM) in both *Tsc2*−/− MEFs and 621-102 AML cells, while some loss of cell viability became apparent using the higher APX2014 concentration of 10 μM (data not shown). Given the limited observed cytotoxicity, the drug activities of APX3330, APX2009, and APX2014 to impair migration, invasion, tumor formation, and transcriptional activity are unlikely through drug cytotoxicity.

### 3.6. Vasculature Mimicry Was Blocked with APX3330, APX2009, and APX2014, While mTOR Inhibition Was Ineffective

During oxidative stress, we hypothesize that Ref-1 activates the STAT3/NF-κB/HIF-1α transcriptional signaling nexus to orchestrate a transcriptional program to drive the mTORC1-independent disease aspects of TSC. Based on the capacity of Ref-1 inhibitors to block STAT3 and HIF-1α activity, we reasoned that Ref-1 inhibition could impact the angiogenesis pathways in TSC. To explore this, we carried out vasculature mimicry assays. *Tsc2*−/− MEFs (Figure 9a) or 621-102 AML cells (Figure 9b) were grown under hypoxia on Matrigel to encourage tube formation. Under hypoxia, both the *Tsc2*−/− MEFs and 621-102 AML cells formed a tubular network that represents the early stages of blood vessel vascularization. Of interest, the inhibition of Ref-1 with APX3330, APX2009, and APX2014 was sufficient to inhibit vasculature mimicry in both the *Tsc2*−/− MEFs and 621-102 AML cells, while mTORC1 inhibition with rapamycin was not effective at the analyzed doses. A significant decrease was observed in the average vessel length after Ref-1 inhibitor treatment with APX3330, APX2009, and APX2014 but not rapamycin (Figure 9a,b). Our data imply that mTORC1 inhibition is not sufficient to block the hypoxia-induced vascularization of these TSC2-deficient cells, while the blockade of redox signal transduction through Ref-1 is effective. This complements and supports what was previously observed in ocular studies and the inhibition of angiogenesis [29,30,31,32]. These data highlight that targeting Ref-1 could possibly have additional benefits to TSC patients by blocking tumor angiogenesis when compared to current therapies with mTORC1 inhibitors.

## 4. Discussion

During conditions of oxidative stress, Ref-1 positively regulates the redox-sensitive transcription factors STAT3, NF-κB, and HIF-1α to orchestrate a transcriptional program to drive angiogenesis, metabolic transformation, and cell migration/invasion. These are critical disease facets of TSC and LAM that are not fully restored with mTOR inhibitors and could help explain why we observe a stable but partial response with mTOR inhibitors in the clinical setting [9]. Therefore, Ref-1 and downstream redox-sensitive transcription factors regulated by Ref-1 present new therapeutic targets. Importantly, our work indicates that Ref-1 inhibition might restore disease aspects of TSC linked to hypoxia and redox imbalance, processes that are less sensitive to mTOR inhibition. Furthermore, our work reveals that Ref-1 inhibition does not induce a selective cytotoxic response in the diseased cells.

We demonstrate that Ref-1 regulated the transcriptional activity of HIF-1α, STAT3, and NF-κB, which are aberrantly elevated in TSC2-deficient cells. Furthermore, Ref-1 inhibition potently blocked the migration and invasion of *TSC2*-deficient cells. It is worth noting that Ref-1 inhibition with APX3330 was more effective than rapamycin to block *Tsc2*−/− MEF cell invasion. This finding indicates that the metastatic traits of TSC2-deficient cells could be dependent on Ref-1 redox signaling involving downstream transcription factors such as STAT3. STAT3 is a key driver of metastasis in many cancers (for a review see [33]). Therefore, our work highlights STAT3 as a possible drug target for the malignant characteristics of TSC-associated LAM (see [6]). The Ref-1/HIF-1α/STAT3/NF-κB signaling node likely contributes to disease attributes linked to TSC pathology, such as inflammation, cell migration/invasion, tumor growth, metabolic transformation, and angiogenesis. The finding that Ref-1 inhibition can simultaneously impact multiple transcription factors makes Ref-1 an attractive drug target for the treatment of TSC.

APX3330 is more than a tool compound, as demonstrated by the results of our phase I APX3330 trial in solid tumors (Clinical Trial Identifier: NCT03375086). We observed a 30% response rate, a RECIST partial response (PR), no significant toxicities, and disease stabilization in six patients, with four on the drug for an extended time (252, 337, 357, and 421 days). We predicted PK and target engagement based on patient biopsies showing decreases in genes regulated by transcription factors regulated by Ref-1, such as HIF-1α, NF-κB, and STAT3 [34,35]. We also observed patient serum levels of APX3330 between 50 and 130 μM. Therefore, the doses presented here are within the achievable levels in humans. APX3330 is also showing a strong safety profile in the ongoing phase 2b diabetic retinopathy (DR)/diabetic macular edema (DME) clinical trial (unpublished data). However, second-generation APX compounds, as shown here, are based on a robust SAR (structure–activity relationship) program, have been identified, and are beginning to be studied in various models, including cancer [26,36], antiocular [32] age-related macular degeneration [37], and inflammatory bowel disease (IBD) models. These studies included both the use of APX2009 and APX2014, which showed greater effects at lower drug concentrations and are nearing pre-IND enabling status.

Metabolic profiling revealed interesting differences between the *Tsc2*−/− MEFs and the wild-type controls. Upon the loss of *Tsc2*, the metabolites involved in glutamine metabolism were markedly different, implicating possible changes to glutamine-dependent purine synthesis, redox homeostasis, and energy metabolism. Glutamate is a precursor for regenerating antioxidant metabolites such as GSH. The shift in the metabolic profile to favor glutamate could have disease implications in the neurons of TSC patients. This is because α-ketoglutarate is transaminated along with glutamine to form the excitatory neurotransmitter glutamate. Indeed, abnormalities in glutamate homeostasis have been linked to seizure frequency in TSC models [38]. Therefore, restoring glutamine/glutamate metabolism with APX3330 and additional analogues might have potential benefits in *TSC2*-deficient cells.

## 5. Conclusions

In TSC2-deficient cells, Ref-1 functions as an important redox sensor that turns on an array of redox-sensitive transcription factors such as STAT3, HIF-1α, and NF-κB. Through these transcription factors, Ref-1 promotes angiogenesis, inflammation, and a metabolic transformation that further supports the tumor growth of TSC2-deficient cells. The activity of Ref-1 is not mediated through mTORC1. Therefore, this Ref-1 redox-sensitive pathway presents itself as a potential therapeutic target to treat disease aspects of TSC that may have additional benefits compared to mTORC1 inhibitors alone.

## Figures and Tables

**Figure 1 cancers-14-06195-f001:**
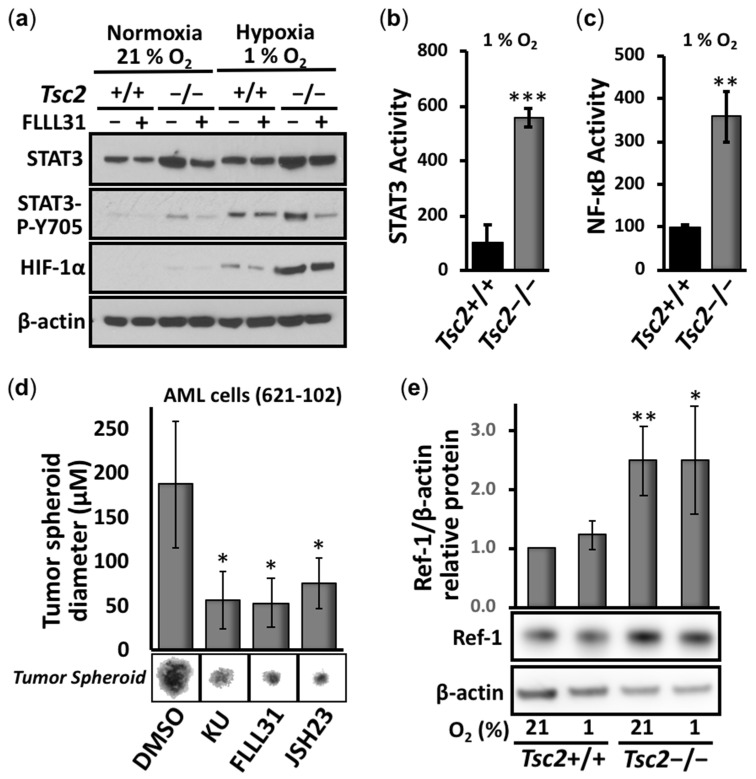
STAT3, HIF-1α, NF-κB, and Ref-1 are possible drug targets in TSC. (**a**) By Western blot, the total STAT3, HIF-1α, β-actin, and P-Y705 STAT3 were determined in *Tsc2*+/+ and *Tsc2*−/− MEFs treated with either DMSO or 5 μM FLLL31 in either 21 or 1% oxygen, where indicated, for 18 h. *Tsc2*+/+ and *Tsc2*−/− MEFs were transiently transfected with (**b**) STAT3 and (**c**) NF-κB luciferase vector and subjected to hypoxia (1% O_2_) for 18 h (*n* = 3). (**c**,**d**) In vitro tumor formation assays were carried out in 621-102 AML cells over 4 weeks in the presence of DMSO, 1 μM Ku-0063794 (KU), 5 μM FLLL31, and 30 μM JSH23. Diameters were measured in Image J (*n* = 30). (**e**) By Western blotting, the relative protein levels of Ref-1 normalized to β-actin were determined in *Tsc2*+/+ and *Tsc2*−/− MEFs in normoxic and hypoxic conditions (21 and 1% oxygen) (*n* = 3). The densitometry of protein bands was measured in Image J. For all graphs, error bars represent standard deviations (SDs); * *p* < 0.05, ** *p* < 0.01, *** *p* < 0.001 compared to DMSO or wild type.

**Figure 2 cancers-14-06195-f002:**
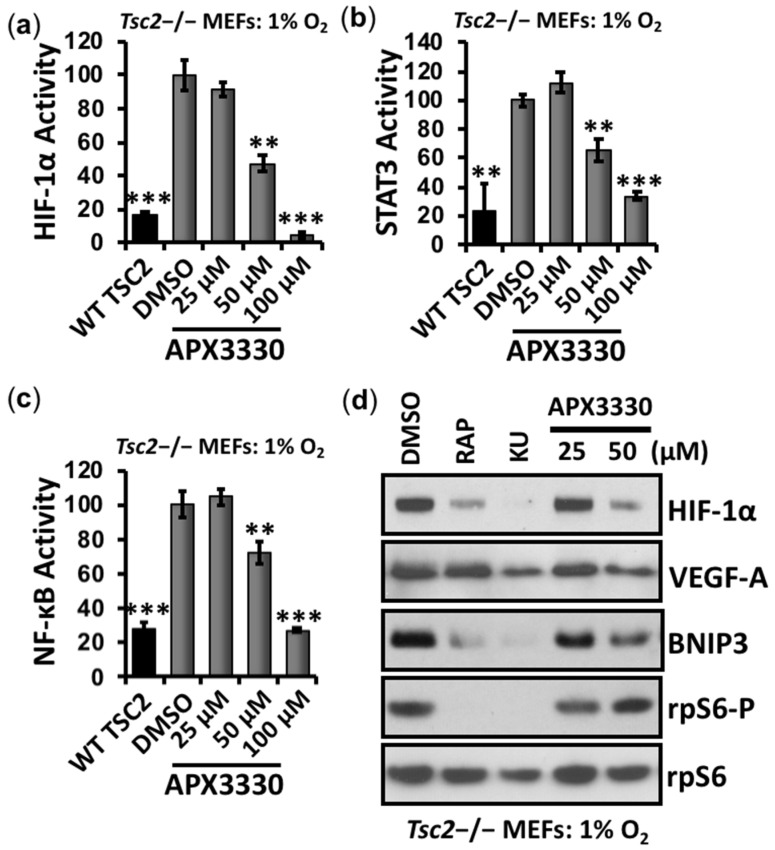
APX3330 inhibits HIF-1α, STAT3, and NF-κB in *Tsc2*-deficient cells. *Tsc2*−/− MEFs were transiently transfected with (**a**) HIF-1α (**b**) STAT3, and (**c**) NF-κB luciferase vector with wild-type TSC2 vector (where indicated). Cells were subjected to hypoxia (1% O_2_) for 18 h in the presence of DMSO or APX3330 (25, 5,0 and 100 μM), and transcription assays carried out, as indicated (*n* = 3). Error bars represent SDs. ** *p* < 0.01, *** *p* < 0.001 compared to DMSO. (**d**) *Tsc2*−/− MEFs were treated with either DMSO, 50 nM rapamycin, 1 μM Ku-0063794 (KU), or APX3330 (25 or 50 μM) for 18 h under hypoxia (1% O_2_). Western blots for total HIF-1α, VEGF-A, BNIP3, and total and phosphorylated rpS6 were carried out (*n* = 3).

**Figure 3 cancers-14-06195-f003:**
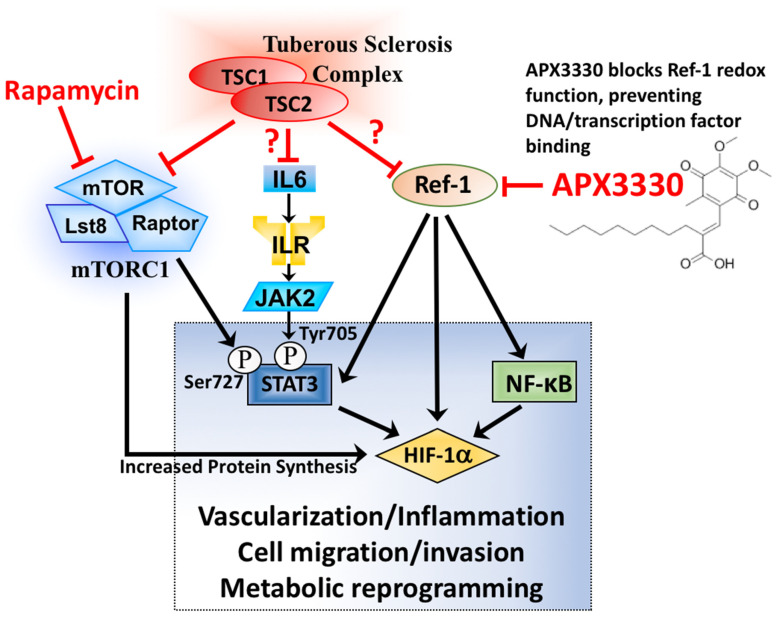
Diagram depicting Ref-1 redox signaling through STAT3, NF-κB, and HIF-1α, a possible drug target for TSC. Multiple signaling inputs from TSC1/TSC2 towards STAT3 and HIF-1α are shown. STAT3, NF-κB, and HIF-1α function as a transcriptional node that orchestrates vascularization, inflammation, cell migration/invasion, and metabolic reprogramming. APX3330 blocks Ref-1′s redox function to prevent the interaction of these transcription factors with DNA.

**Figure 4 cancers-14-06195-f004:**
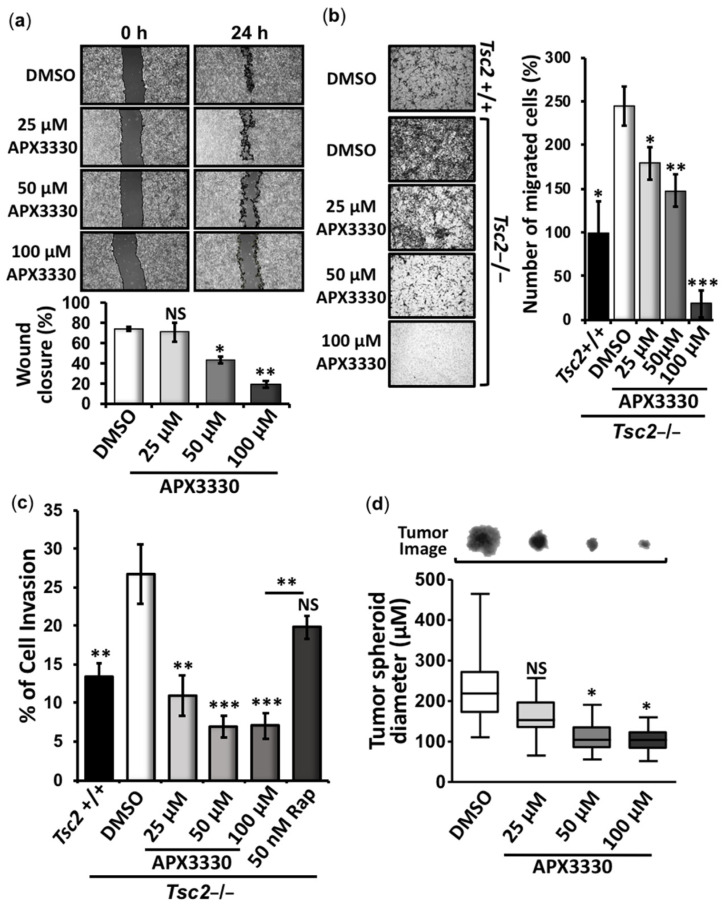
APX3330 inhibits the cell migration/invasion and tumor formation of *Tsc2*−/− MEFs. *Tsc2*−/− MEFs were subjected to (**a**) wound scratch assays (*n* = 3), (**b**) cell migration (*n* = 3), (**c**) cell invasion (*n* = 4), and (**d**) in vitro tumor formation assays (30 tumors analyzed within three biological repeats) in the presence or absence of either DMSO or APX3330 (25, 50, and 100 μM), where indicated. *Tsc2*+/+ MEFs were used as controls in both the cell migration and cell invasion assays. Error bars represent SDs (SEMs for (**d**)); * *p* < 0.05, ** *p* < 0.01, *** *p* < 0.001 when compared to DMSO.

**Figure 5 cancers-14-06195-f005:**
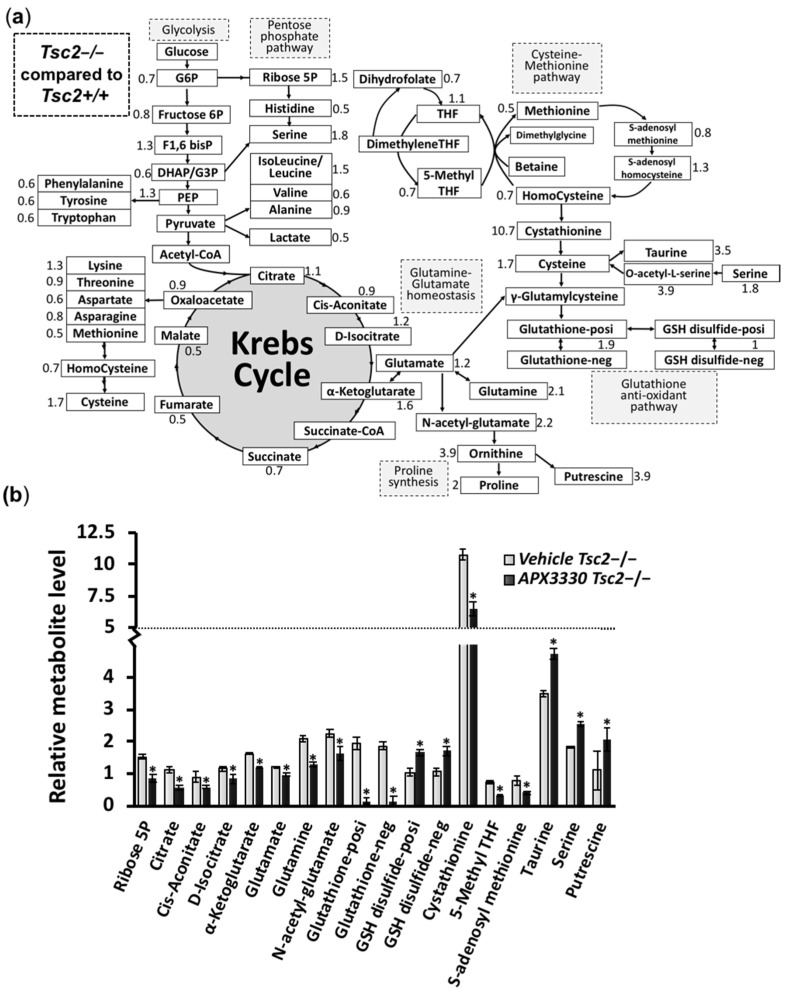
Metabolic profile of *Tsc2*−/− MEFs treated with APX3330 compared to *Tsc2*+/+ MEFs. (**a**) *Tsc2*+/+ and *Tsc2*−/− treated for 18 h with either DMSO or 25 μM APX3330 in low-serum conditions (0.1% (*v*/*v*) FBS) were subjected to metabolomics (*n* = 5). Metabolites are depicted and include glycolysis, the pentose phosphate pathway, the Krebs cycle, the cysteine–methionine pathway, glutamine–glutamate homeostasis, proline synthesis, and the glutathione antioxidant pathway. The values outside the metabolite boxes are the mean fold-change of the *Tsc2*−/− MEFs when compared to the *Tsc2*+/+ MEFs (where a value of 1 is equal between cells). Changes >1.5-fold are underlined, and cystathionine is labeled with an * as the metabolite with the highest difference. Blackened boxes are metabolites that are reduced by APX3330. (**b**) Top scoring metabolites that are modulated by AX3330 are graphed, showing fold changes of untreated and APX3330-treated *Tsc2*−/− MEFs normalized to untreated *Tsc2*+/+ MEFs. Error bars represent SDs; * *p* < 0.05, ** *p* < 0.01, *** *p* < 0.001 when comparing untreated versus APX3330 in Tsc2−/− MEFs.

**Figure 6 cancers-14-06195-f006:**
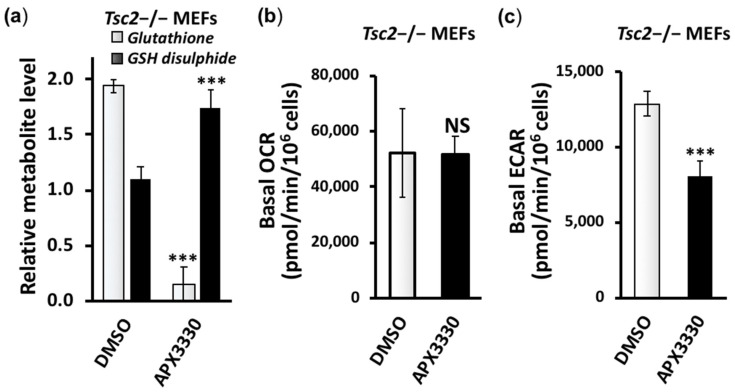
APX3330 treatment affects glutathione homeostasis and ECAR but not OCR. (**a**) The levels of glutathione and glutathione disulfide in the untreated and 25 μM APX3330-treated *Tsc2*−/− MEFs are graphed. (**b**) Basal oxygen consumption rate (OCR) and (**c**) extracellular acidification rate (ECAR) assessed in *Tsc2−/−* MEFs after 18 h treatment of 25 μM APX3330 in 10% (*v*/*v*) FBS compared to DMSO. Error bars represent SDs; *** *p* < 0.001 when compared to their paired DMSO control.

**Figure 7 cancers-14-06195-f007:**
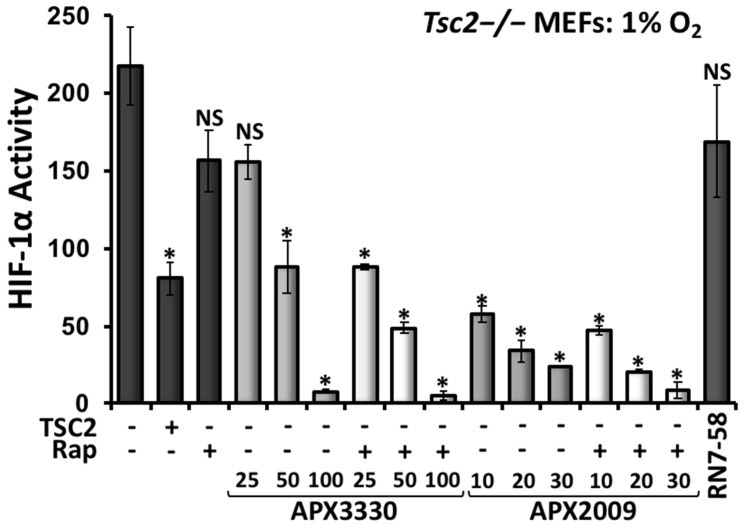
The second-generation Ref-1 inhibitor APX2009 potently inhibited HIF-1α, while rapamycin was less effective. *Tsc2*−/− MEFs were transiently transfected with the HIF-1α luciferase vector (and the TSC2/pcDNA3.1 vector as a control, where indicated). These cells were subjected to hypoxia (1% O_2_) for 18 h in the presence of DMSO, 50 nM rapamycin, APX3330 (25, 50 and 100 μM), or APX2009 (10, 20, and 30 μM), or 100 μM RN7-58 (control drug) as single agents, or where indicated, rapamycin was combined with APX3330 and APX2009. Luciferase assays to determine HIF-1α activity were carried out (*n* = 3). Error bars represent SDs; * *p* < 0.05, ** *p* < 0.01, *** *p* < 0.001 when compared to the untreated DMSO control.

**Figure 8 cancers-14-06195-f008:**
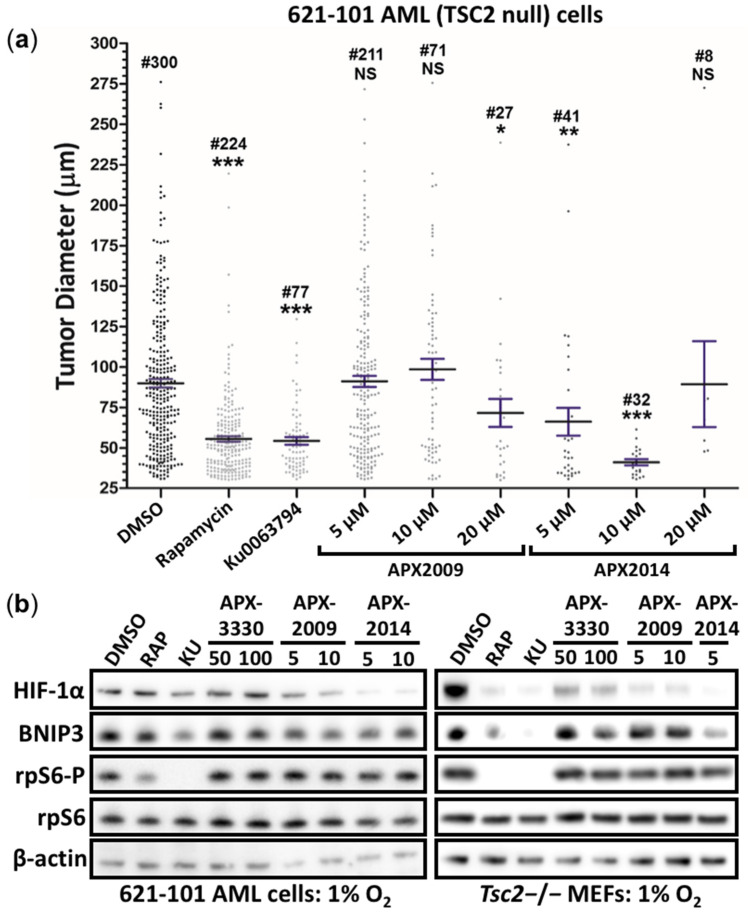
APX2009 and APX2014 show potency to inhibit in vitro tumor growth of 621-102 AML cells. (**a**) The 621-102 AML cells were subjected to in vitro tumor formation assays in the presence or absence of DMSO, 50 nM rapamycin, 1 μM Ku0063794, APX2009 (5, 10, and 20 μM), or APX2014 (5, 10, and 20 μM), where indicated. Tumor diameters were quantified; the number of tumors (#) are indicated above each column (up to 100 tumors analyzed within each of three biological repeats). # refers to the number of tumors that were present. Error bars represent SEMs; * *p* < 0.05, ** *p* < 0.01, *** *p* < 0.001 when compared to the untreated DMSO control. (**b**) The 621-102 AML cells (left panel) or *Tsc2*−/− MEFs (right panel) were treated with DMSO, 50 nM rapamycin, 1 μM Ku-0063794 (KU), APX3330 (50 or 100 μM), APX2009 (5 or 10 μM), or APX2014 (5 or 10 μM), where indicated, for 18 h under hypoxia (1% O_2_). From prepared lysates, Western blots for total HIF-1α, BNIP3, total and phosphorylated rpS6, and β-actin were carried out (*n* = 3).

**Figure 9 cancers-14-06195-f009:**
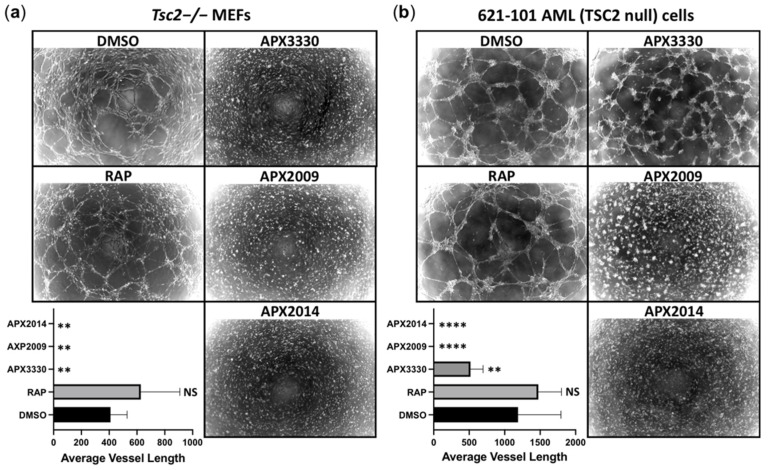
Ref-1 inhibitors blocked vasculature mimicry, while rapamycin was ineffective. Vasculature mimicry assays were carried out on (**a**) *Tsc2*−/− MEFs and (**b**) 621-102 AML cells on Matrigel under hypoxia (1% O_2_) in the presence of DMSO, 50 nM rapamycin, 50 μM APX3330, 5 μM APX2009, or 5 uM APX2014 (*n* = 5). The average vessel length was quantified using AngioTool. A one-way ANOVA with Tukey’s multiple comparisons was carried out; not significant (NS), ** *p* < 0.01, **** *p* < 0.0001 when compared to untreated DMSO.

## Data Availability

The data presented in this study are available in this article (and Appendix A).

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
