# Peer review of "Drug Inhibition of Redox Factor-1 Restores Hypoxia-Driven Changes in Tuberous Sclerosis Complex 2 Deficient Cells"

_cancers, 2022, doi:10.3390/cancers14246195_

Round 1

Reviewer 1 Report

Dear authors

The work is interesting since it explores new therapeutic targets for TSC.

Comments

Lines 105-117. The objective of the study is sufficient for the readers. The results obtained should be given in the section results. 

Subsection 2.. A reference should be aded

The diagramm 2e should be further detailed and presented in a separate figure since it can be of interest for the readers 

Author Response

Lines 105-117. The objective of the study is sufficient for the readers. The results obtained should be given in the section results. 

Response:We have removed any indication of the results from the end of the introduction.

Subsection 2.. A reference should be added

Response:We had 3 references in the Material and Methods section where required. We do not see where another reference could be added. We opted to go for a more fuller description of the methods to reduce the necessity of using references.

The diagramm 2e should be further detailed and presented in a separate figure since it can be of interest for the readers 

Response:A new diagram in color showing the complexity and multifaceted regulation of NF-kB, STAT3 and HIF-1alpha is presented, with mTORC1 and IL6/JAK2 directed input as well as the Ref-1 redox signalling. Thank you for this suggestion.

Reviewer 2 Report

Champion at el. submitted a well-designed study verifying the role of Ref-1 in TSC-2 deficient cells and its involvement in the regulation of STAT3 and NF-kB activity and its downstream elements, especially HIF-1a. The authors used TSC2-/- cells and metabolic inhibitors to verify their hypothesis about the regulatory role of TSC2 in the metabolic adaptation of tumor growth. The results are well presented, and all controls are included. The workflow is smooth, and the following experiments support each other. No major concerns appeared during the article revision.

Minor issues:

1.   Please, include the antibodies concentration in the Materials and Methods section.

2.   For all reagents, the company, city, state (if relevant) and country should be provided. Please update.

3.   In graphs (2A,B,C, 3D), the space between the number and unit should be added.

4.   References 12, 13, 19, 22 - capital letters should be used: Mol. Cancer Res.

5.   References 19, 22,22, 2424 - abbreviated journal - abbreviated journal title should be used

Author Response

Champion at el. submitted a well-designed study verifying the role of Ref-1 in TSC-2 deficient cells and its involvement in the regulation of STAT3 and NF-kB activity and its downstream elements, especially HIF-1a. The authors used TSC2-/- cells and metabolic inhibitors to verify their hypothesis about the regulatory role of TSC2 in the metabolic adaptation of tumor growth. The results are well presented, and all controls are included. The workflow is smooth, and the following experiments support each other. No major concerns appeared during the article revision. 

Response: Thank you for your comments. The minor issues have been addressed below.

Minor issues:

  1. Please, include the antibodies concentration in the Materials and Methods section.

Response:Antibody concentrations were now added to the Material and Method section.

  1. For all reagents, the company, city, state (if relevant) and country should be provided. Please update.

Response:The Material and Method section has been updated to now include this information.

  1. In graphs (2A,B,C, 3D), the space between the number and unit should be added.

Response:This formatting has been now correctly amended in the graphs.

  1. References 12, 13, 19, 22 - capital letters should be used: Mol. Cancer Res.
  2. References 19, 22,22, 2424 - abbreviated journal - abbreviated journal title should be used

Response for 4 and 5: The references have been reformatted for Cancers. Thank you for noticing these corrections.

Reviewer 3 Report

In this research, Champion et al investigated the role of Ref-1 in the treatment of Tuberous sclerosis. In the TSC2-deficient cells, inhibition of Ref-1 prohibited hyperactivity of STAT3, NF-kb, and HIF-1a that are known to be regulated by Ref-1 gene.  When the Ref-1 inhibitors were administered to TSC-/- MEF, the three genes (STAT3, NF-kb, and HIF-1a) were down-regulated proportional to the concentration of the inhibitors. In addition, inhibition of vasculature mimicry was observed with the administration of the inhibitors. 

This research is interesting, and the results seem to be meaningful to the development of non-mTOR therapeutics for tuberous sclerosis. However, I have some concerns about the research. 

Major

1. The main problem of this research is lack of evidences about chemotherapeutic potentials of the Ref-1 inhibitors to TC. In vitro tumor growth experiments revealed limited cytotoxicity of the Ref-1 inhibitors, therefore it is hard to accept the future utility of the inhibitors in the treatment of TC. 

2. In the metabolomics experiments, GSH:GSH disulphide ratio changed dramatically in the TSC2-/- MEFs. This indicates that reactive oxygen status changed aberrantly in the cells, which implicates relatedness to apoptotic signaling. I think the apoptotic status should be checked in addition to the cytotoxicity. 

3. Although the research is well-designed and results are interesting, it seems to be lack of novel findings. In fact, the three genes are known to be regulated by Ref-1. Therefore, it is quite obvious that the genes are regulated according to the gene expression status of the Ref-1. Of course, it should be identified experimentally, but the novelty of the expected finding is relatively low. If additional pathways were explored, the results would have more novelty. 

4. I'd like to know why the authors didn't use the TC cell line instead of MEF. In the experiment of potential therapeutics, it is natural to apply the materials to cell line or animal models. If those materials had been used, the results would have better implications. 

Minor

1. A full description of MEF is required. (Mouse Embryonic Fibroblast?)

2. In Fig. 1, each sub-figures should be annotated in lowercase. 

3. Page 6, line 245 -> '>' symbols needs to be changed to '<' ("> 0.05*"  ->  "< 0.05*")

Author Response

In this research, Champion et al investigated the role of Ref-1 in the treatment of Tuberous sclerosis. In the TSC2-deficient cells, inhibition of Ref-1 prohibited hyperactivity of STAT3, NF-kb, and HIF-1a that are known to be regulated by Ref-1 gene.  When the Ref-1 inhibitors were administered to TSC-/- MEF, the three genes (STAT3, NF-kb, and HIF-1a) were down-regulated proportional to the concentration of the inhibitors. In addition, inhibition of vasculature mimicry was observed with the administration of the inhibitors. 

This research is interesting, and the results seem to be meaningful to the development of non-mTOR therapeutics for tuberous sclerosis. However, I have some concerns about the research. 

Response:thank you for your comments. We have addressed the major comments below, and agree that future development of the work presented in this article is warranted in other cancer types and drug combinations as well as to advance this work into in vivomodels to bridge the gap towards clinical application in patients.

Major

  1. The main problem of this research is lack of evidences about chemotherapeutic potentials of the Ref-1 inhibitors to TC. In vitro tumor growth experiments revealed limited cytotoxicity of the Ref-1 inhibitors, therefore it is hard to accept the future utility of the inhibitors in the treatment of TC. 

Response:We agree that chemotherapy combination with Ref-1 inhibitors is an important consideration when treating sporadic cancers, to enhance current chemotherapies that already exist for cancer patients. The issue with Tuberous Sclerosis Complex patients, is that they have acquired germline mutation of either TSC1or TSC2in all their cells (unless it is a mosaicism), making them extremely vulnerable to DNA damaging (genotoxic) therapies. DNA damage of their only good copy of TSC1or TSC2would lead to the development of a tumor. This is evident, as UV-induced DNA damage is known to cause skin tumors (angiofibromas) in TSC patients. Therefore, we don’t think there is a strong enough rational for exploring chemotherapies in combination with Ref-1 inhibitors for treatment specifically in TSC patients. However, the indication of Ref-1 inhibitors in other non-TSC cancer therapies is definitely something that will be pursued in future studies.

  1. In the metabolomics experiments, GSH:GSH disulphide ratio changed dramatically in the TSC2-/- MEFs. This indicates that reactive oxygen status changed aberrantly in the cells, which implicates relatedness to apoptotic signaling. I think the apoptotic status should be checked in addition to the cytotoxicity. 

Response:We agree that the changes in the GSH:GSH disulphide ratio would typically result in cells to undergo cell death. However, analysis of cell death through the assays already presented in this manuscript (as supplementary data) show that the TSC2-deficient cells (in 2 different cell line models) are still able to survive, with only a low level of cell death at only the higher Ref-1 inhibitor concentrations used.

We did carry out additional experiments, work not presented in this manuscript’s. Western blot analysis from cell extracts of cells treated with Ref-1 inhibitors did not reveal the detection of standardly used apoptotic markers (PARP cleavage and/or Caspase cleavage). As we utilised a suitable methodology to quantify cell death already presented in supplementary, and saw no appreciable cell death, we think that the cell death experiments already presented in the manuscript should suffice for the readership.

The question of why these TSC2-deficient cells survive is a question that will need much further investigation, and is its own story in itself. We speculate that the high tolerance to the GSH:GSH disulphide ratio change is due in part to the high level of anti-oxidant capacity that these TSC2-deficient cells already possess before treatment with the Ref-1 inhibitors. For another manuscript we are preparing to publish, we carried out RNA sequencing and found high expression of anti-oxidant gene-sets that would protect these TSC2-deficient cells to oxidative cell stress. We also have data showing that TSC2-deficient cells are markedly resistant to a panel of agents that induce oxidative cell stress. We are finishing off these experiments for this other paper. When we are ready to submit this other paper, we will cross reference to this Cancerspaper.

  1. Although the research is well-designed and results are interesting, it seems to be lack of novel findings. In fact, the three genes are known to be regulated by Ref-1. Therefore, it is quite obvious that the genes are regulated according to the gene expression status of the Ref-1. Of course, it should be identified experimentally, but the novelty of the expected finding is relatively low. If additional pathways were explored, the results would have more novelty. 

Response:This paper is the first indication that Ref-1 could be involved in TSC, and presents itself as an mTOR-independent signaling mechanisms of disease. Everything presented in this paper is novel as it relates to TSC. The action of Ref-1 inhibitors to block vascular mimicry in TSC cells is especially novel, as vascular mimicry has never been examined before in the context of TSC, and the lack of efficacy of mTOR inhibitors to block vascular mimicry is a surprising result. I believe the way the paper was written (especially with the old mechanistic figure), it would appear that everything is logical and targeting Ref-1 is an obvious approach. In reality though, transcriptional hyperactivity of NFkB, STAT3 and HIF-1alpha is complex with many inputs that could be derived from loss of TSC2. This includes mTOR hyperactivity and a higher level of inflammatory signaling through IL6 and JAK2 towards STAT3. mTOR is known to increase the translation of HIF-1alpha protein and promote HIF-1alpha transcription activity. mTORC1 also phosphorylates STAT3 to enhance activity directly. IL6/JAK2 hyperactive signaling also leads to STAT3 phosphorylation and further activation. Given this complexity, finding a therapeutic strategy to effectively repress all three NFkB, STAT3 and HIF-1alpha transcription factors is a technical challenge. We hope that providing a fuller mechanistic figure as suggested by reviewer 1 (as Figure 3) will help get this idea across to the readership. The potency of Ref-1 to restore the disease state that is mTORC1-independent in TSC is particularly exciting, and should be of considerable interest to TSC researcher and clinicians.

  1. I'd like to know why the authors didn't use the TC cell line instead of MEF. In the experiment of potential therapeutics, it is natural to apply the materials to cell line or animal models. If those materials had been used, the results would have better implications. 

Response:This paper was an in vitroanalysis of drug action and potential. We utilized at least 2 cell line models in this manuscript as is expected for a publication in Cancers. We agree that other pre-clinical models of TSC should be tested with Ref-1 inhibitors, but this was out of the project costs that was funded by TSC charity money. This manuscript would help support the rational to acquire additional funds to test Ref-1 inhibitors within in vivomodels of TSC.

Minor

  1. A full description of MEF is required. (Mouse Embryonic Fibroblast?)
  2. In Fig. 1, each sub-figures should be annotated in lowercase. 
  3. Page 6, line 245 -> '>' symbols needs to be changed to '<' ("> 0.05*"  ->  "< 0.05*")

Response:we have addressed all these minor comments in the corrected manuscript

Reviewer 4 Report

The manuscript has a sound methodology and the results are reported well. I have no major comment for this article. A minor comment:

It has been mentioned that the ddCT (delta-delta-Ct) method was used and normalized to a single control gene (β-actin)” However, accurate normalization of real-time quantitative RT-PCR data by geometric averaging of “multiple” internal control genes.

Author Response

The purpose of Figure 1E was to show over-expression of Ref-1 in cells lacking TSC2. We did this by examining the relative levels of Ref-1 protein and mRNA. We agree that using only one reference/house-keeping gene is not optimal for RNA expression analysis by q-PCR. We originally carried out the mRNA analysis by q-PCR 3-4 years ago for Figure 1E, during a time when using only one reference gene was standard practice for these types of experiments. After discussing options with the current research team, we feel that perhaps showing the q-PCR mRNA expression data is not the most optimal experiment to quantify Ref-1 expression. It would be much better if we had instead quantified total Ref-1 protein levels, rather than mRNA.
            Therefore, we felt it would improve the manuscript if we instead provided n=3 expression blots of Ref-1 protein with quantification by densitometry. We have updated figure 1E The results and conclusions are the same as originally presented in Figure 1E during review. We have amended the material and methods and figure legends.

Round 2

Reviewer 3 Report

The authors' responses to reviewer's comments are acceptable.